# The efficacy of Netrod Six-electrode Radiofrequency Renal Denervation system: mathematical modeling, phantom experiments and animal studies

Xiaohua Song[1,2‡], Jiulin Guo[3‡], Hongguang Cao[3], Qun Nan [1]*

**1** College of Chemistry and Life Science, Beijing University of Technology, Beijing, China, **2** Industrial Informatization and Science and Technology Bureau of Changqing District, Jinan, China, **3** Shanghai Golden Leaf MedTech CO., Ltd, Shanghai, China,

‡ These authors are joint senior authors on this work.
* nanqun@bjut.edu.cn

## Abstract

The principle of Netrod™ Renal Denervation RF Generator is based on the heating of radiofrequency electrodes to achieve the therapeutic temperature for peripheral nerve ablation with minimal impact on intra-arterial blood flow. The extravascular heat distribution and continuous ablation were evaluated using COMSOL Multiphysics with an idealized 3D model and validated in an ex vivo experiment. The safety of ablation was verified by H&E staining of a hypertensive dog model in the treatment group of animal experiments, and changes in blood pressure before and after the procedure(assessed at 7, 30, and 60 days in animal experiments) were used to characterize the efficacy. The simulation results showed that the maximum injury depth range of the Netrod™ Six-electrode is 5.08mm and the depth of the ablation zone obtained in the Phantom experiment was 3–6mm. Circumferential coverage was up to 91.99% and the ablation zone was continuous. In animal experiments, eight animals in the treatment group had significantly lower blood pressure (88.9%). Nerve cells in nerve bundles showed vacuolar changes and thickening of peripheral sheaths after peripheral nerve ablation of renal arteries.The Netrod™ Six-electrode presented here has potential to effectively and safely ablate renal nerves. The demonstrated efficacy and safety in animal models provide a foundation for clinical trials in humans.

## Introduction

Hypertension is a cardiovascular syndrome with elevated arterial pressure in the body circulation as the main clinical manifestation and is an important risk factor for cardiovascular and cerebrovascular diseases. Hypertension is a common chronic disease that continues to increase in prevalence worldwide and is expected to reach 1.56 billion people by 2025, with refractory hypertension(RH) being particularly

**Data availability statement:** All relevant data are within the paper.

**Funding:** This work was supported by the National Natural Science Foundation of China [31771021, 11832003]. The funder had role in study design, data collection and analysis.

**Competing interests:** The authors have declared that no competing interests exist.

challenging to treat. [1]. Refractory hypertension(RH) is defined as hypertension in which blood pressure cannot be achieved even after adequate and reasonable combination of at least 3 antihypertensive drugs (including diuretics) based on dietary modification and lifestyle improvement [2]. RH accounts for about 5–15% of hypertensive patients, and some sources report that it can reach up to 20%-30%. There is still a clinical need for new easy, safe and effective treatments for hypertension.

Excessive sympathetic activation plays a significant role in the development and maintenance of hypertension [3–5]. Several studies have indicated that blocking sympathetic nerves may serve as a viable target for hypertension treatment [6–8]. Renal denervation (RDN) is a minimally invasive procedure that reduces blood pressure by modulating sympathetic hyperexcitability [9]. This technique is gaining acceptance and widespread application in clinical settings across various countries [10].Other studies [11] have also shown that RDN has the potential to extend to other autonomic nervous system disorders such as atrial fibrillation [12,13], heart failure [14,15] and tachycardia [16].

Although research related to RDN is still in its early stages, several new devices have been developed in recent years [17]. Common radiofrequency (RF) catheter ablation devices include the Symplicity System, Symplicity Spyral System, EnligHTN System, FlashPoint™ System, and Iberis System. We have developed the Netrod™ Six-electrode Radiofrequency Renal Denervation system and have investigated the safety and efficacy of the system in simulations, phantom experiment, and animal studies.

In the following sections, we will detail the numerical simulations and validation experiments conducted with the Netrod™ Six-electrode. This research aims to provide a scientific basis for assessing the clinical efficacy of renal sympathetic nerve ablation using the Netrod™ Six-electrode.

## Methods

### Equipment introduction

Netrod™ Six-electrode Radiofrequency Renal Denervation system (Shanghai Golden Leaf MedTech Co. Ltd) includes Netrod™ Renal Denervation RF Generator and Netrod™ Six-electrode Radiofrequency Renal Denervation catheter (Fig 1).

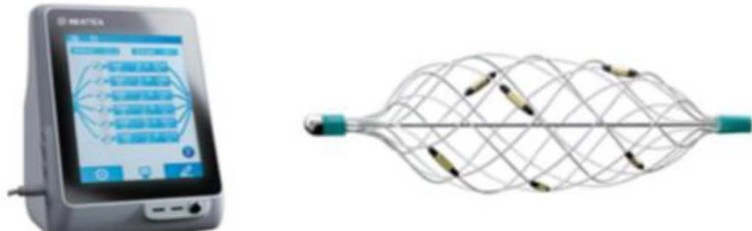

**Fig 1. Radiofrequency ablation mainframe and Netrod™ Six-electrode structure diagram.**

The generator primarily consists of a microcontroller, a signal source circuit, a power drive circuit, a power output circuit, a detection circuit, and a display operation control unit. The detection circuit mainly detects monitors voltage, current, and temperature levels (Fig 2).

The microcontroller is the core component of the generator. It not only processes the voltage, current, and temperature data transmitted by the detection circuit, but also outputs impedance, temperature, and power data to the display operation control unit. Furthermore, it intelligently adjusts the power output of the signal sources based on temperature and impedance to maintain the electrode surface temperature at a constant value, thereby ensuring safety during operation. The display operation control unit real-time records and displays the temperature, impedance, and power of each electrode during the radiofrequency ablation, and draws relevant curves (Fig 3), allowing medical staff to monitor the working status of the system in real-time through the display.

The radiofrequency ablation catheter is composed of a central wire drawing with a developing head, a mesh tube connector, an electrode, a mesh basket bracket, a catheter sleeve, a seven-hole tube, and a handle assembly (Fig 4).The simulated schematic of the radiofrequency ablation catheter entering the vessel is shown in Fig 5.

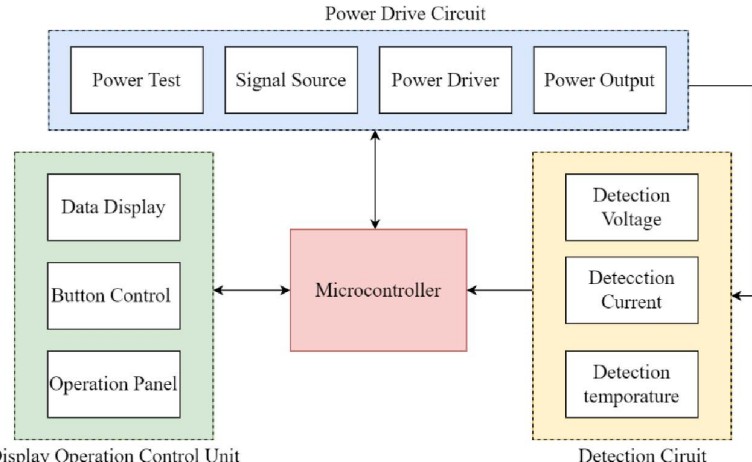

**Fig 2. Netrod™ Renal Denervation RF Generator components.**

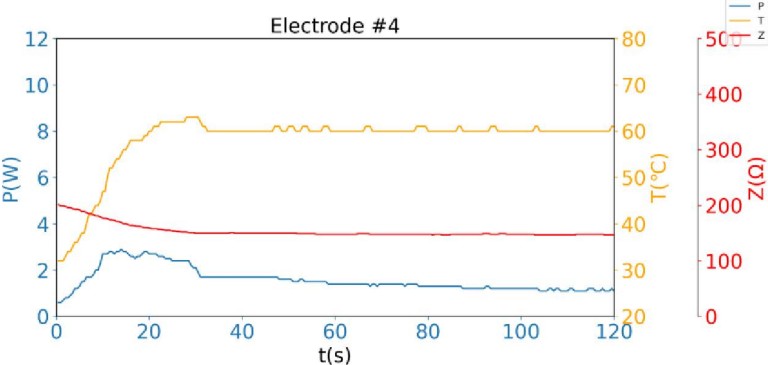

**Fig 3. 120 seconds of radiofrequency ablation, the power(P), temperature(T), and impedance(Z) of electrode 4 change over time.**

                                                  

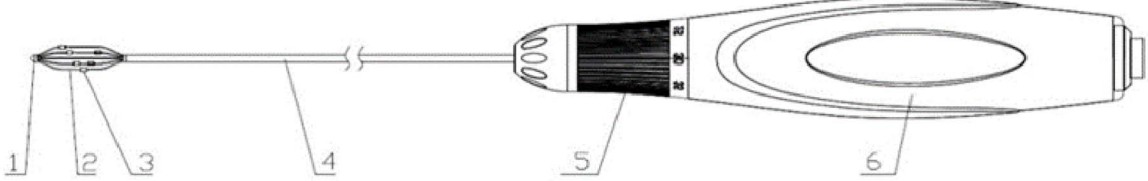

**Fig 4. Netrod™ Six-electrode Basket Radiofrequency Renal Denervation Catheter.** 1: Central wire drawing with a developing head; 2: Mesh basket bracket; 3: Electrode; 4: Seven-hole tube; 5: Adjustment knob; 6: Handle.

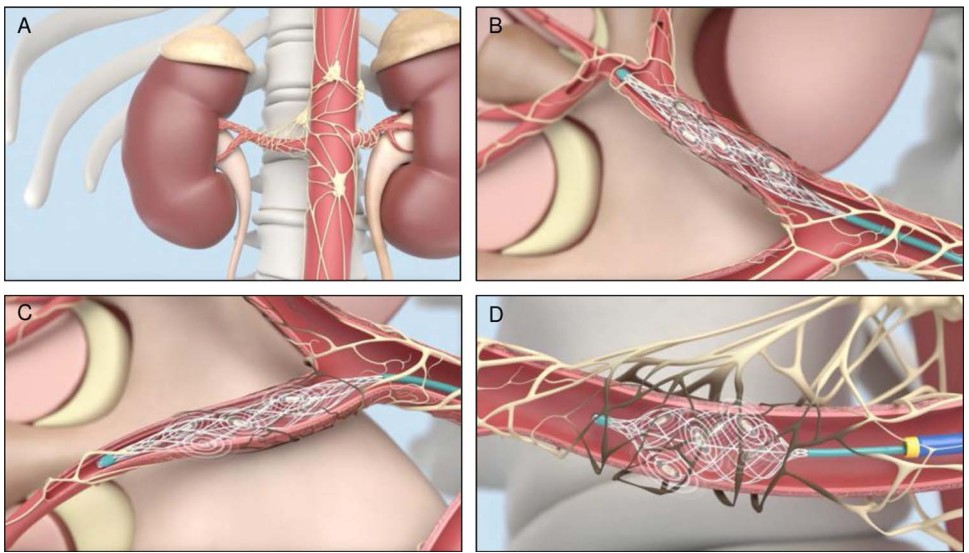

**Fig 5. Basket catheter ablation of renal sympathetic nerves.** A. Renal sympathetic nerve distribution; B,C. The basket catheter enters the branch for ablation; D. The basket catheter expands the basket stent to perform ablation in the main trunk of the renal artery.

### Finite element method modeling

We use COMSOL Multiphysics software (COMSOL, Burlington MA, USA), a finite element simulation software, to perform electromagnetic field and temperature field coupling calculations.

In radiofrequency ablation therapy, thermal energy is generated by converting the electrical energy that is applied as the application of radiofrequency current to the target renal artery tissue. The electric field control equation [18] can be calculated by solving the following Laplace equation. The frequency of the RF current is set to 470 KHz [19], and the biological medium can be considered resistive because the displacement current is much smaller than the conduction current and thus can be neglected [20]. The distributed heat source Qr is given by Eq. (1)

$$Q_r = J \cdot E \tag{1}$$

Where J is the current density, measured in A/m²; Qr is the distributed heat source, measured in J. The voltage is obtained from the solved Laplace equation, which is the controlling equation for solving the electric field problem in the presence of the internal power supply; see Eq. (2) and (3).

$$\nabla \cdot (\sigma \nabla V) = 0 \tag{2}$$

$$E = -\nabla V \tag{3}$$

Where V is the voltage, measured in V; E is the electric field strength, measured in V/m; $\sigma$ is the electrical conductivity, measured in S/m. In the model, the boundary conditions at the current interface are set as follows:

$V = 0$ All external surfaces of the abdominal cavity

$$V = V_0 \text{ Six electrodes outer surface} \tag{4}$$

$$\boldsymbol{n} \cdot J = 0 \text{ All other domain boundaries}$$

The initial potential is set to 0 V, the outer surface of the electrodes is set to a voltage value with temperature control. In the entire model, all the outer surfaces of the tissue are grounded with zero current density.

In the simulation, an idealized three-dimensional vascular geometry model of the renal artery was constructed, as shown in Fig 6. The tissue included two major parts, the renal artery (6mm diameter) and the abdominal cavity (26mm×26mm×30mm), which was constructed as the external environment for the simulation. The arterial inlet flow rate was set to 0.5 m/s, and the ablation time was set to 120s. The material property parameters of the abdominal portion, blood, and electrodes are shown in Table 1.

Where $\rho$ is the density, kg/m³; σ is the conductivity, S/m; C is the specific heat capacity, J/kg·°C; K is the thermal conductivity, W/(m °C); $\xi_r$ is the relative dielectric constant.

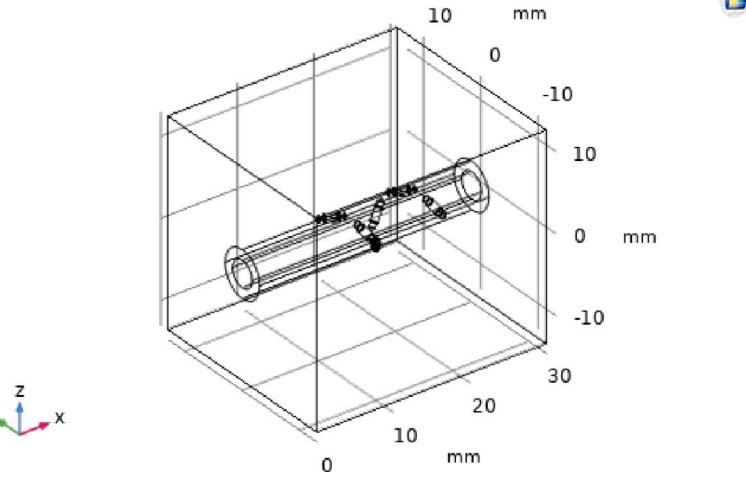

**Fig 6. Three-dimensional renal artery tissue and net basket-based RF electrode model.**

**Table 1. shows material property parameters of the abdominal part, blood and electrodes [21–25].**

| Material Properties | $\rho$(kg/m³) | σ (S/m) | C(J/kg °C) | K (W/(m °C) | $\xi_r$ |
|---|---|---|---|---|---|
| Blood | 1050 | 0.99 | 4180 | 0.54 | 58.3 |
| Abdominal cavity | 1060 | 1.69 | 3600 | 0.512 | 43.03 |
| Agar | 1300 | 2.3 | 4182 | 0.6 | 57.99 |
| Gold Electrode | 19300 | 4.1e7 | 127 | 315 | 1 |

## Phantom experiment validation

Tissue-mimicking phantoms play a crucial role in the development, performance testing, quality assurance, and formulation of effective treatment plans for RF ablation devices [26]. Given that the walls of real human renal vessels are very thin, the wall thickness is often neglected in experimental models. In this study, we utilized a cylindrical mold with a vascular channel, where the surrounding area simulates human abdominal tissue. The entrance diameter of the vascular channel is 6 mm, corresponding to the dimensions of the simulation model. Polyacrylamide is particularly advantageous due to its stability at high temperatures. Many researchers have reported successful applications of polyacrylamide-based transparent tissue-mimicking molds, incorporating protein as a temperature indicator for thermal ablation studies [27]. The raw materials and formulation details for the specific mold used in this study are presented in Table 2. [28]

The target temperature for the electrode was set at 60°C, and the experiment was conducted for a duration of 120 seconds. For thermal damage analysis, we evaluated the depth of thermal damage in the body mold through post-experimental assessment. The body mold changes color from blue to white when the temperature exceeds 45°C. Therefore, the color change on the surface of the body mold served as an indicator of the ablation effect.

## Animal study

Dogs subjected to animal testing and experimental protocols were treated in strict accordance with protocols approved by the Institutional Animal Care Committee of Shanghai Harborside Medical Technology Co., Ltd., based on the Institutional Animal Care and Use Committees (IACUC) Animal Care and Use Program on Animal Experimentation (approval No. IACUC-2021–003).

A total of 18 female experimental dogs (treatment group 101–109 and control group 201–209, 9 animals each) that had successfully established hypertension models were included in this experiment and divided into three points of time, with 6 dogs in each point of time (3 animals each in the treatment group and the control group). The three points of time were 7 days (101–103 for the treatment group and 201–203 for the control group), 30 days (104–106 for the treatment group and 204–206 for the control group) and 60 days (107–109 for the treatment group and 207–209 for the control group). Under X-ray fluoroscopy, the left and right renal arteries of the animals in the treatment group were ablated with the system, and the operational performance of the experimental dogs in the treatment group was evaluated, and the Digital Subtraction Angiography (DSA) images are shown in Fig 7(a)–7(d). The duration of ablation was set to 120s. As a measure of efficacy, blood pressure was measured at regular intervals before and during the post-test observation period, and we measured the change in blood pressure at three points of time after RDN. During survival, animals were observed clinically. To investigate the short-term, medium-term, and long-term safety of the experimental dogs in the treatment group, three points of time, 7, 30, and 60 days, were included in this experiment. On postoperative days 7 (±1), 30 (±3) and 60 (±5) (with the

**Table 2. Proportioning table for the preparation of 100 ml solution [29].**

| Substance Name | Weighing |
|---|---|
| Citric acid | 2.49g |
| Sodium citrate | 2.66g |
| Acrylamide | 7.62g |
| Methylene-bisacrylamide | 0.4g |
| Bovine serum protein | 2.0g |
| Degassed water | 100ml |
| Ascorbic acid | 0.1g |
| 1%$FeSO_4$ | 1.0ml |
| 3%$H_2O_2$ | 1.0ml |

day of surgery as 0 day), renal arteriograms were performed in animals scheduled for dissection to visualize the test site and renal arteries with nerves were harvested for immuno-histochemical evaluation. Animals were euthanized by deep anesthesia and an overdose of potassium chloride for autopsy and histopathological analysis. Quantification of protein denaturation and necrosis of renal sympathetic nerves around renal arteries was performed using hematoxylin-eosin (H&E) staining.

## Results

### Simulation results of Netrod™ Six-electrode

As the ablation time increases, the ablation area expands, with data detailed in Table 3.

With prolonged ablation time, the boundary of the region where the temperature reaches 45°C expands. Since temperature-controlled RF ablation was employed, the maximum temperature was confined to the vicinity of the electrodes, as shown in Table 3. The Netrod™ Six-electrode device can reach temperatures up to 45°C at distances of approximately 5.08 mm (>4 mm) from the electrode.

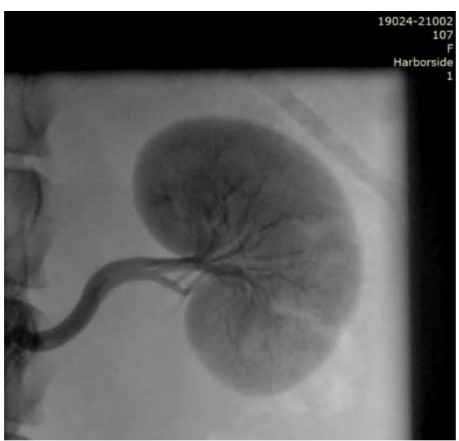

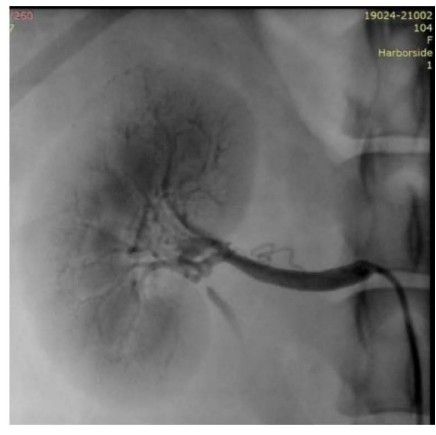

a.   Left renal artery angiography(Animal 107 in the treatment group )

b. Right renal artery angiography(Animal 104 in the treatment group )

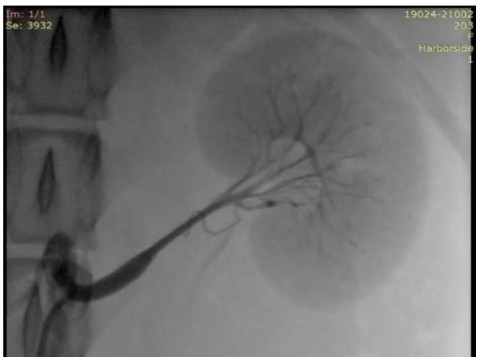

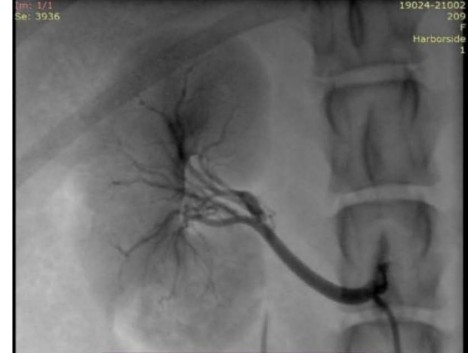

c.Left renal artery angiography(Animal 203 in the control group)

d. Right renal artery angiography(Animal 209 in the control group )

**Fig 7. Typical images of preanatomic renal arteriogram.**

As shown in Fig 8, The Netrod™ Six-electrode can work simultaneously, and the mesh basket structure allows the electrodes to deliver RF energy close to the vessel wall. The Netrod™ Six-electrode can achieve 360° total ablation of sympathetic nerves in 30s.

The continuity of the Netrod™ Six-electrode ablation zone at an ablation time of 30s is shown in Fig 9. Each of the Netrod™ Six-electrode sits on a separate mesh basket of wires, which better conforms to the vessel wall as the lumen changes, keeping the electrode arrangement consistent.

## Results of phantom experiments

As shown in Fig 10, during the ablation process, the six electrodes discharge to form a continuous 360° spiral ribbon, with the ablation ranges of the first and sixth electrodes overlapping radially. The RF ablation experiments resulted in an ablation zone with a width of 6–8 mm and a depth of 3–6 mm, as depicted in Fig 10.

The results from the phantom experiments validate the continuous ablation zone of the Netrod™ Six-electrode observed in the simulation. These experiments demonstrate that, despite the discrepancies in depth, the width of the ablation zone remains consistent, further supporting the accuracy of the simulation model. By comparing the body model experiments with the simulation results, we can confirm the effectiveness of the six electrodes in achieving a 360° spiral ablation. This consistency not only enhances confidence in the simulation model but also provides a reliable basis for clinical applications, indicating that the electrodes can effectively control ablation range and depth during actual procedures.

**Table 3. Ablation of electrodes at different times.**

|  | Electrode temperature/˚C | Distance between 45˚C boundary and electrodes/mm |
|---|---|---|
| 5s | 48.6 | 0.23 |
| 30s | 60.0 | 2.69 |
| 60s | 60.0 | 3.78 |
| 120s | 60.0 | 5.08 |

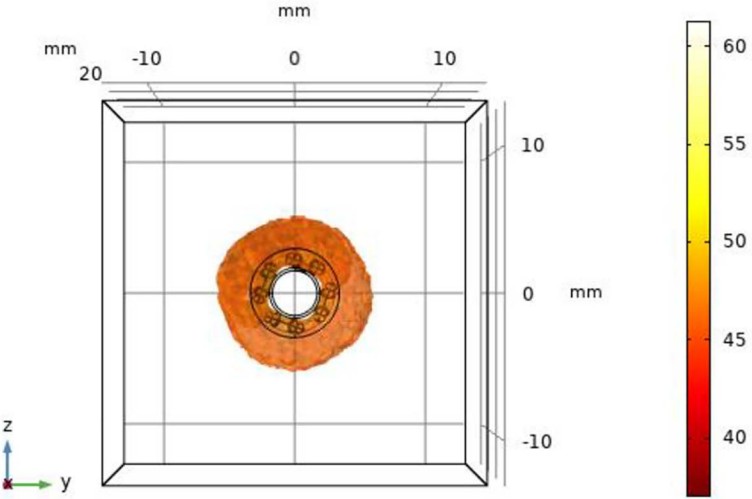

**Fig 8. The effective ablation angle of Netrod™ Six-electrode(t = 30s).**

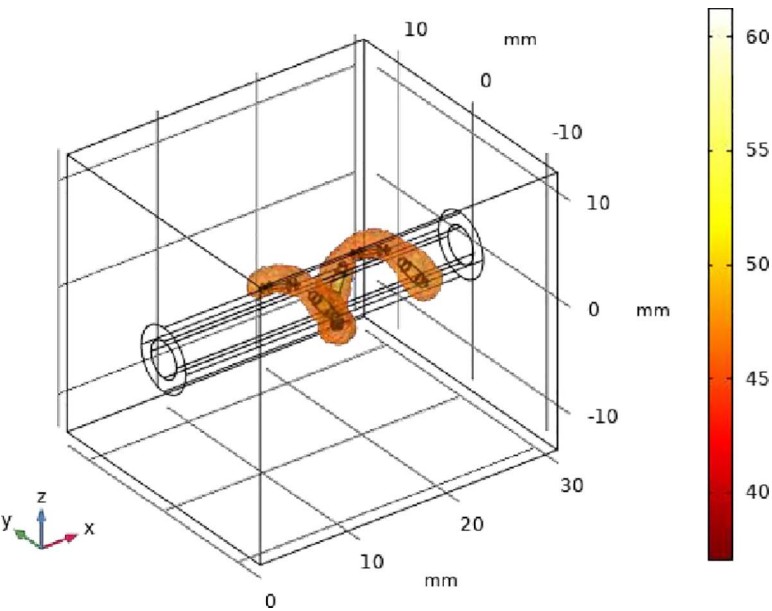

**Fig 9. The continuity of the ablation area between the Netrod™ Six-electrode(t = 30s).**

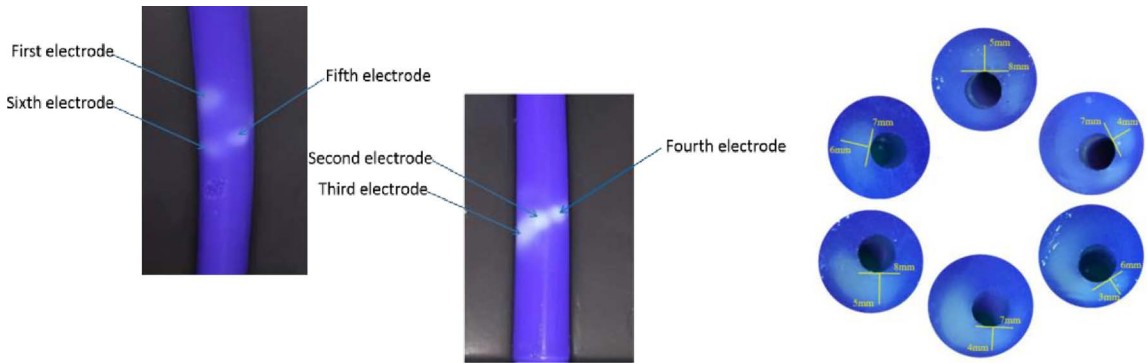

**Fig 10. Ablation effect of body mold experiment.**

## Animal experiment results

The surgeon performed radiofrequency ablation of the left and right renal arteries of the hypertensive model animals according to the protocol (9 animals in the treatment group) or a sham procedure with renal artery angiography only (9 animals in the control group). There were no intraoperative abnormalities or serious adverse events related to animal vital signs, and the operability of the system was rated as excellent by the surgeon.

The group of 18 experimental animals was mentioned in the previous section. The blood pressure of the animals was measured before and periodically after the procedure, and the blood pressure reduction data at different points of time are shown in Fig 11. Of the three animals in the test group at the point of time of 7 (±1) days, post-ablation diastolic and mean arterial pressures in animal 101 may not be statistically different relative to pre-ablation due to individual differences (P ≥ 0.05). In the other two animals, systolic, diastolic, and mean arterial pressure were significantly lower than before

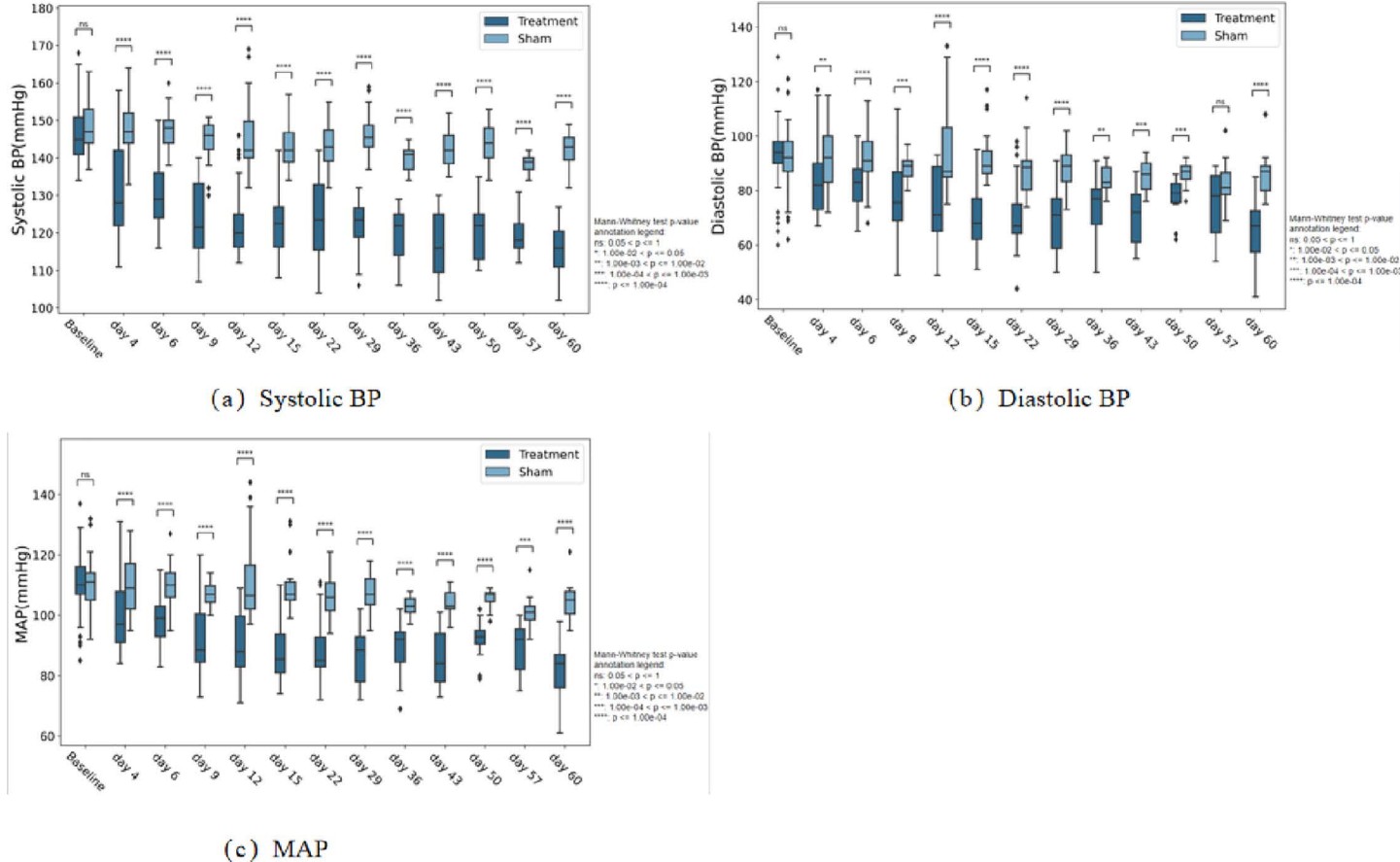

**Fig 11. Values of blood pressure reduction in animals of the treatment group at different point of time.**

ablation (P ≤ 0.05). Blood pressure was significantly lower than before ablation (P ≤ 0.05) in all six animals in the test group at the 30 (±3) and 60 (±5) day point of time. Pre-ablation systolic, diastolic, and mean arterial pressures in the nine control animals during this procedure were only marginally reduced or increased relative to pre-sham procedure, with no significant differences (P ≥ 0.05).

The body weights of the experimental animals were not significantly abnormal during the experimental period, and the clinicopathological indices of the animals did not show any significant health-related abnormalities in the preoperative and pre-dissection test results. During the postoperative observation period, the experimental animals did not show any abnormalities related to the treatment. No abnormalities such as stenosis or dissection were observed in the renal arteries of the animals when the angiography was performed before dissection, and no abnormalities were observed when the animals were dissected for observation and sampling. The above experiments prove the safety of the device. The results of the validity analysis in the histopathological analysis showed that nerve fiber damage around bilateral renal artery vessels in animals of the treatment group at each point of time, showing different degrees of vacuolar degeneration, hyaline degeneration and coagulative necrosis. Degeneration of nerve fibers around the renal artery vessels was also seen in pathological section samples from most of the control animals, which was considered to be a nerve injury reaction caused by the sham operation itself. In the histopathological analysis, we measured the maximum distance from the point of ablation to the nerve fiber injury in animals in the treatment group. The ablation depth measurement was performed by a professional animal pathologist to identify the damaged nerve fibers furthest from the ablation location, which was

automatically measured by pathology image software, and the measured depth is shown schematically in Fig 12. The distribution of ablation depth data is shown in Fig 13.

The distance of nerve injury at the three points of time was not particularly different, and it was inferred that reversible repair of the injured nerve would not occur over time. The results of H&E staining of the ablation sites showed that the intima of the renal artery vessels in all groups of animals showed varying degrees of thickening and inflammatory infiltration. None of them showed desquamation, dissection, calcification or organization. Occasionally, there is thrombus formation in the lumen, which was considered to be caused by blood clotting in the blood vessels during animal dissection. Vascular injury and inflammatory infiltration of the tunica media were present to varying degrees in all groups of animals and no contractures or pseudoaneurysm were present. Occasional hyaline degeneration of the muscularis was seen in both groups, which was considered to be caused by ablation or the procedure itself. Different degrees of necrosis of perivascular tissue, inflammatory infiltration, fibrosis and fatty infiltration were seen in the adventitia and perivascular tissue of renal arteries in both groups, and the overall pathological irritant response was more pronounced in the test group than in the control group. An overview of the efficacy and safety index pathological results is shown in Fig 14 for representative samples at different time periods, with animal number 101 (7 days), 104 (30 days) and 107 (60 days) for the treatment group and 201 (7 days), 204 (30 days) and 207 (60 days) for the control group.

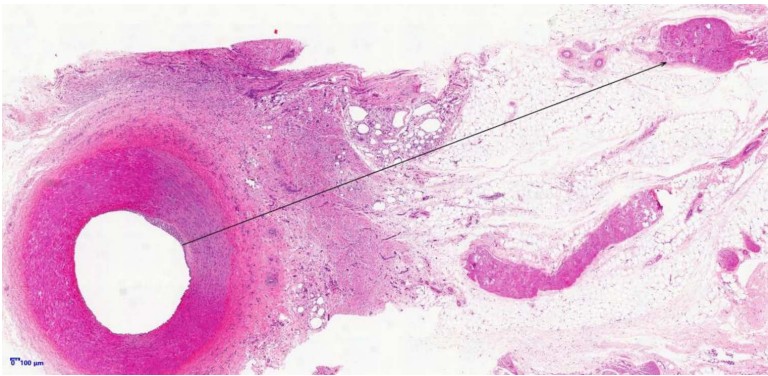

**Fig 12. Measurement depth schematic.**

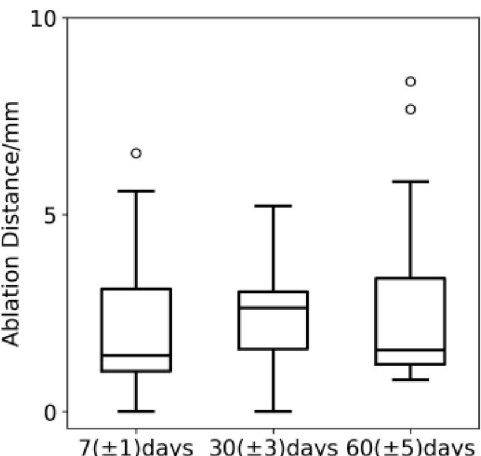

**Fig 13. Distance of nerve damage at different points of time.**

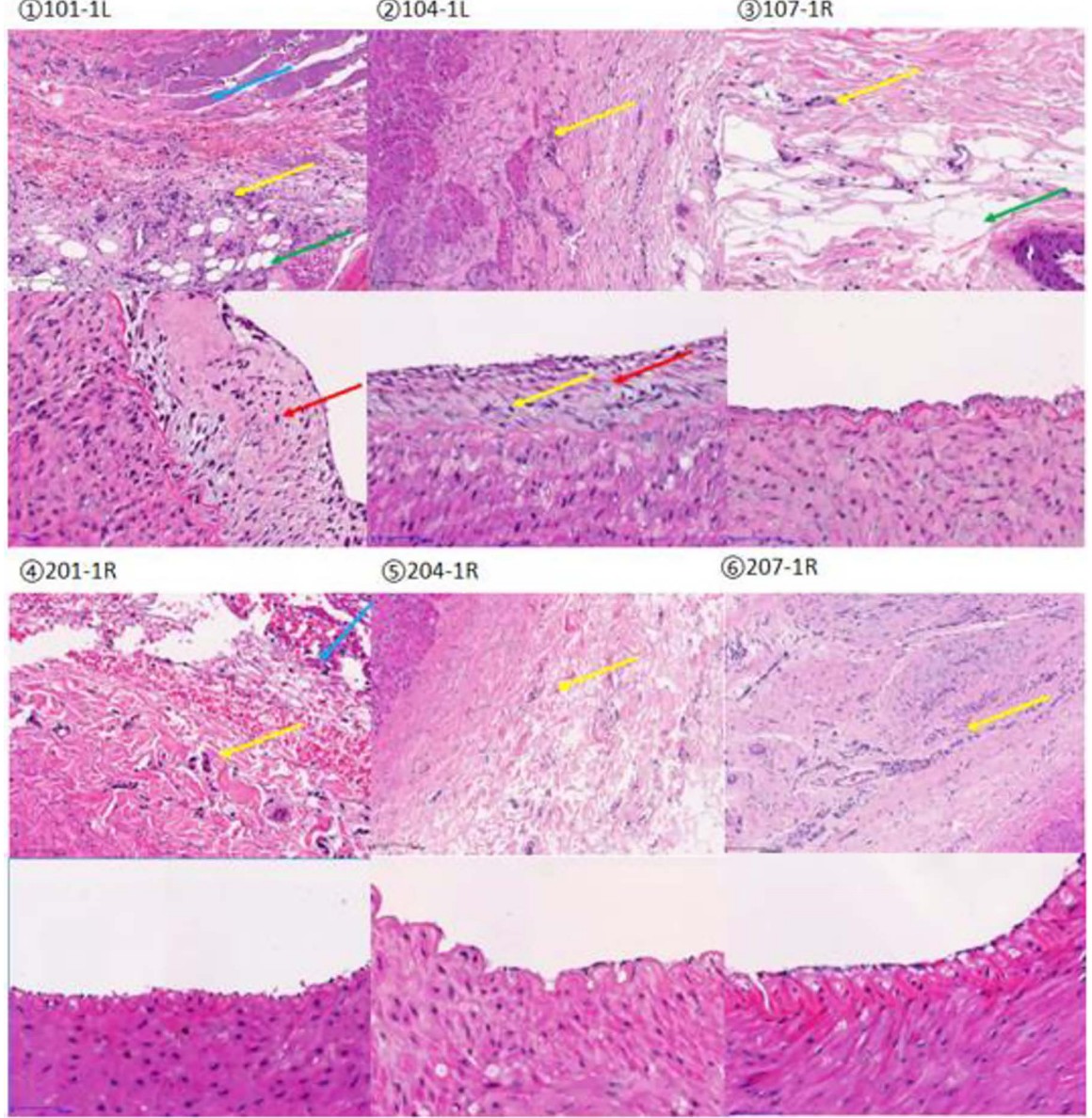

**Fig 14. H&E staining of nerves (inflammatory cells shown by yellow arrows; tissue necrosis shown by blue arrows; fatty infiltration shown by green arrows; intimal thickening shown by red arrows).**

Fig 15A and 15B show the peripheral nerve of the renal artery after ablation, and a large number of vacuolar changes of nerve cells in the nerve bundle are seen; C and D are the peripheral nerve of the renal artery in the control group, and no significant abnormal changes were seen in the nerve bundles. The H&E staining pathological pictures of the canine renal artery peripheral nerve 42 days after the controlled experimental radiofrequency ablation of the renal artery are shown in Fig 16. The peripheral nerve bundle of the renal artery in the test group showed morphological changes and thickening of the peripheral sheath. No significant abnormal changes were seen in the peripheral nerve bundle of the renal artery in the control group. Combining the pathological results of the above efficacy and safety indicators, it was inferred

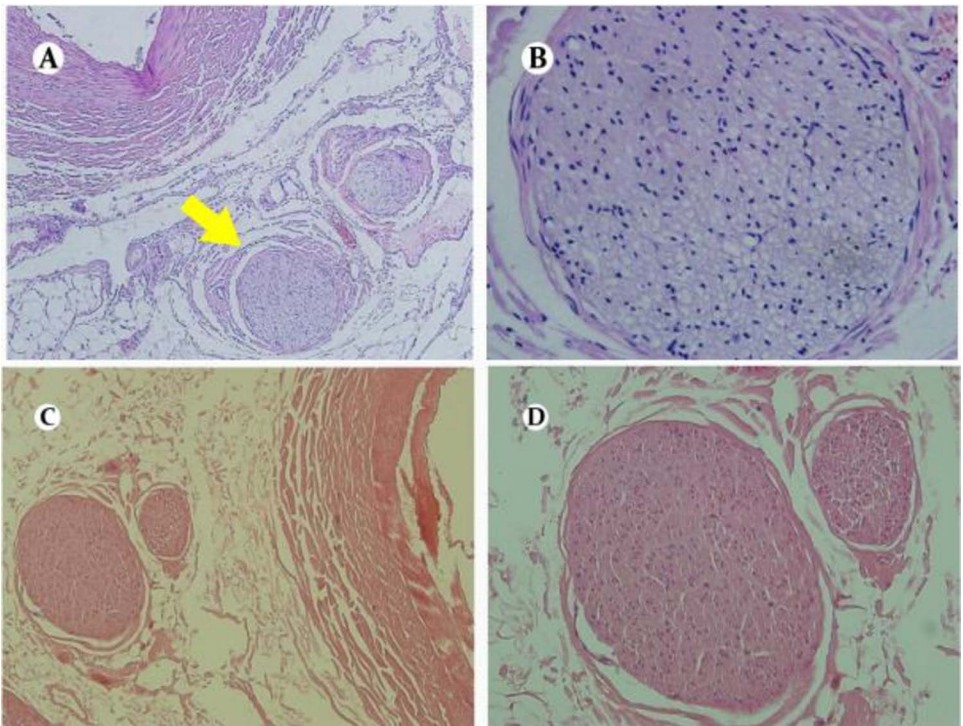

**Fig 15. H&E staining of the peripheral nerve of the renal artery.**

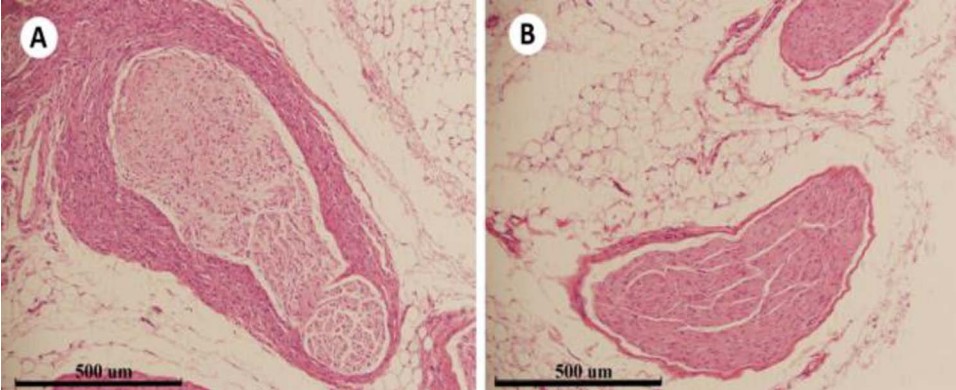

**Fig 16. H&E staining of the peripheral nerve of the renal artery after 42 days.**

that the treatment could effectively ablate perivascular nerve tissue and cause irreversible damage to nerve fibers. The degree of local inflammation and other pathological reactions in the treatment group was significantly greater than that in the control group at all points of time, which was considered to be caused by intraoperative ablation damage to tissues, but did not show significant safety problems, and all of them decreased with the prolongation of postoperative time.

## Discussion

The simulation results showed that the maximum injury depth range of the Netrod™ Six-electrode is 5.08 mm and the depth of the ablation zone obtained in the phantom experiment was 3–6 mm. Circumferential coverage was up to 91.99% and the ablation zone was continuous. In animal experiments, eight animals in the treatment group had significantly lower blood pressure (88.9%). Nerve cells in nerve bundles showed vacuolar changes and thickening of peripheral sheaths after peripheral nerve ablation of renal arteries. The Netrod™ Six-electrode presented here has potential to effectively and safely ablate renal nerves.

The study reports a significant reduction in blood pressure in 88.9% of the treated animals, highlighting the potential efficacy of the Netrod™ system in managing hypertension. Renal denervation has emerged as a promising therapeutic option. The findings suggest that the Netrod™ system could offer a safe and effective treatment option for patients with resistant hypertension who do not respond adequately to pharmacological therapies. The demonstrated efficacy and safety in animal models provide a strong foundation for clinical trials in humans. If successful, this technology could expand treatment options for hypertension and potentially other conditions influenced by sympathetic overactivity, such as heart failure or chronic kidney disease. The research findings on the Netrod™ Renal Denervation RF Generator and its Six-electrode system highlight its potential as a safe and effective tool for renal denervation. Further clinical studies are warranted to validate these findings in human patients and explore broader applications in cardiovascular and renal diseases.

Al Raisi et al. [30] reported a circumferential coverage angle of 54.90°±4.36° for a single-electrode RF application, encompassing approximately 15% of the arterial wall circumference. In contrast, the circumferential coverage angle that can be achieved in the Netrod™ six-electrode simulation experiment is 360°. Yanyan Cheng et al. noted that the placement of spiral electrodes could cause blood to flow spirally along the catheter, potentially resulting in more significant blood flow disturbances. Following balloon electrode placement in the renal artery, normal blood flow is obstructed, resulting in maximum blood flow velocity at the front of the electrode [31]. The mesh basket design of the Netrod™ Six-electrode effectively mitigates blood flow obstruction caused by both the spiral structure and the balloon.

The relevant parameters (including size and material) of the electrodes used in the ex vivo and animal experiments are the same as those of the electrodes used in the simulation. However, our preliminary research has some limitations. First, we utilized a simplified model of the Netrod™ Six-electrode and the renal arteries, omitting the full structure of the Netrod™ Six-electrode and only simulating the electrode positions. Although the electrode positions can vary and adhere to the vessel wall according to the lumen diameter, the contact force between the electrodes and tissue was not considered in our model. While simulation studies provide valuable insights, there remains a significant gap between these findings and their clinical application. In a clinical setting, factors such as patient-specific anatomy, variability in tissue properties, and real-time physiological responses can significantly influence the outcomes of RF ablation procedures. Secondly, the dynamic nature of blood flow and its interaction with the ablation device must be carefully assessed to optimize treatment effectiveness and minimize complications. Additionally, the animal experiments were selected from 18 female dogs, and it was not possible to determine whether gender had an effect on the results of the experiment. And experimental dogs may not reflect the characteristics of human arteries with hypertension. Depth data on the canine model does not fully reflect ablation depths in other animals and in humans. Finally, in this study, we verified the effectiveness of this system through a modeling study, and the animal experiment was a further step to verify the effectiveness and safety of this system, as the data in the modeling study was set up in an ideal state, and there was a gap between the depth of the animal experiment and the animal experiment due to the real arterial structure of the animals and the shape of the arteries. The depth measured in animal studies was based on the furthest nerve damaged, rather than soft tissue involved. Hence, the depth of ablation could be underestimated inherently, if the distribution of nerves surrounding the renal artery segment was scarce beyond 3 mm and this as one of the reasons why there were discrepancies between the depths observed in different settings.

To bridge this gap, future studies should focus on integrating more complex and realistic models that account for these variables, including variations in blood flow dynamics and tissue response during ablation. Clinical trials will also be essential to validate the efficacy and safety of the Netrod™ Six-electrode in diverse patient population demographics, ensuring that laboratory findings translate effectively into real-world applications.

## Conclusion

Through mathematical modeling, phantom experiments and animal study, we established a novel renal denervation system based on the Netrod™ Six-electrode. This system utilizes automatic temperature control and a mesh basket structure to target the nerves around the renal arteries. Simulation studies of the Netrod™ Six-electrode demonstrated that heat distribution effectively achieved denervation of the renal sympathetic nerves while maintaining a constant electrode temperature of 60°C.

This paper focuses on mathematical modeling, phantom experiments and animal studies to demonstrate the feasibility of the proposed new technique. We performed renal denervation in hypertensive dogs model and in an open surgical setting to show the efficacy and safety of the system in vivo. Further clinical studies are needed to validate the effectiveness and safety of the NetrodTM Six-electrode denervation system for blood pressure control in vivo.

## Author contributions

**Data curation:** Jiulin Guo, Hongguang Cao.

**Funding acquisition:** Qun Nan.

**Software:** Xiaohua Song.

**Writing – original draft:** Xiaohua Song.

**Writing – review & editing:** Qun Nan.

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
