## [Decision Letter · Decision Letter 0]

26 Nov 2024

PONE-D-24-46135Study on the efficacy of NetrodTM Six-electrode Radiofrequency Renal Denervation system on renal artery vesselsPLOS ONE

Dear Dr. Nan,

Thank you for submitting your manuscript to PLOS ONE. After careful consideration, we feel that it has merit but does not fully meet PLOS ONE’s publication criteria as it currently stands. Therefore, we invite you to submit a revised version of the manuscript that addresses the points raised during the review process.

This is an interesting study that requires work. Though the study is novel, it requires the addressing of all comments raised by both the reviewers.

Additionally, in the first paragraph of Introduction, I would mention the clinical utility of renal denervation currently, beyond hypertension. You could refer to reviews encapsulating these findings, such as: https://doi.org/10.1016/j.cpcardiol.2023.102196

Furthermore, the discussion requires work. An additional paragraph on comparing and contrasting your findings with existing literature, what's novel about your results, and some background with pros and cons on all three types of electrodes used and their comparisons in either the introduction or second paragraph of the discussion is warranted. 

We look forward to receiving your revised manuscript.

Kind regards,

Academic Editor

PLOS ONE

Journal Requirements: When submitting your revision, we need you to address these additional requirements. 1. Please ensure that your manuscript meets PLOS ONE's style requirements, including those for file naming. The PLOS ONE style templates can be found at https://journals.plos.org/plosone/s/file?id=wjVg/PLOSOne_formatting_sample_main_body.pdf and https://journals.plos.org/plosone/s/file?id=ba62/PLOSOne_formatting_sample_title_authors_affiliations.pdf 2. Please note that PLOS ONE has specific guidelines on code sharing for submissions in which author-generated code underpins the findings in the manuscript. In these cases, we expect all author-generated code to be made available without restrictions upon publication of the work. Please review our guidelines at https://journals.plos.org/plosone/s/materials-and-software-sharing#loc-sharing-code and ensure that your code is shared in a way that follows best practice and facilitates reproducibility and reuse. 3. Thank you for stating the following financial disclosure: "This work was supported by the National Natural Science Foundation of China [31771021, 11832003]."  Please state what role the funders took in the study.  If the funders had no role, please state: ""The funders had no role in study design, data collection and analysis, decision to publish, or preparation of the manuscript."" If this statement is not correct you must amend it as needed. Please include this amended Role of Funder statement in your cover letter; we will change the online submission form on your behalf. 4. We note that your Data Availability Statement is currently as follows: All relevant data are within the manuscript and its Supporting Information files. Please confirm at this time whether or not your submission contains all raw data required to replicate the results of your study. Authors must share the “minimal data set” for their submission. PLOS defines the minimal data set to consist of the data required to replicate all study findings reported in the article, as well as related metadata and methods (https://journals.plos.org/plosone/s/data-availability#loc-minimal-data-set-definition). For example, authors should submit the following data: - The values behind the means, standard deviations and other measures reported;- The values used to build graphs;- The points extracted from images for analysis. Authors do not need to submit their entire data set if only a portion of the data was used in the reported study. If your submission does not contain these data, please either upload them as Supporting Information files or deposit them to a stable, public repository and provide us with the relevant URLs, DOIs, or accession numbers. For a list of recommended repositories, please see https://journals.plos.org/plosone/s/recommended-repositories. If there are ethical or legal restrictions on sharing a de-identified data set, please explain them in detail (e.g., data contain potentially sensitive information, data are owned by a third-party organization, etc.) and who has imposed them (e.g., an ethics committee). Please also provide contact information for a data access committee, ethics committee, or other institutional body to which data requests may be sent. If data are owned by a third party, please indicate how others may request data access. 5. Please amend either the abstract on the online submission form (via Edit Submission) or the abstract in the manuscript so that they are identical. 6. Please ensure that you refer to Figure 5 in your text as, if accepted, production will need this reference to link the reader to the figure.

**Additional Editor Comments:**

This is an interesting study that requires work. Though the study is novel, it requires the addressing of all comments raised by both the reviewers.

Additionally, in the first paragraph of Introduction, I would mention the clinical utility of renal denervation currently, beyond hypertension. You could refer to reviews encapsulating these findings, such as: https://doi.org/10.1016/j.cpcardiol.2023.102196

Furthermore, the discussion requires work. An additional paragraph on comparing and contrasting your findings with existing literature, what's novel about your results, and some background with pros and cons on all three types of electrodes used and their comparisons in either the introduction or second paragraph of the discussion is warranted.

Reviewers' comments:

Reviewer's Responses to Questions

**Comments to the Author**

1. Is the manuscript technically sound, and do the data support the conclusions?

Reviewer #1: Partly

Reviewer #2: No

2. Has the statistical analysis been performed appropriately and rigorously? 

Reviewer #1: N/A

Reviewer #2: N/A

3. Have the authors made all data underlying the findings in their manuscript fully available?

Reviewer #1: Yes

Reviewer #2: No

4. Is the manuscript presented in an intelligible fashion and written in standard English?

Reviewer #1: No

Reviewer #2: Yes

5. Review Comments to the Author

Reviewer #1: In this manuscript, the authors examined the depth and circumference of ablation lesions produced by the Netrod and other radiofrequency renal denervation systems through simulation models and phantom experiments. While the information provided has clinical significance, there are several methodological limitations and content inconsistencies that cast doubt on the conclusions.

The title (referring to efficacy and safety, as shown on the first page, which differs from the title on the first page of the text) is inconsistent with the content presented, which only addresses the extent of ablations and does not discuss efficacy or safety.

Major limitations requiring clarification:

Abstract Inconsistency: The abstract on the first page of the PDF file mentions animal studies and histological analysis. However, the main text and its abstract only present data on ablation lesion depth and circumference from different radiofrequency renal denervation systems. Additionally, the titles differ between sections. The content is limited and inconsistent with the title on the first page ("Study on the efficacy and safety of Netrod™ Six-electrode Radiofrequency Renal Denervation system on renal artery vessels"), as no information regarding clinical efficacy and safety is presented.

Unsupported Assumptions in Introduction: The authors assume that "continuous circumferential (ring)" ablation is superior for renal denervation without providing supporting evidence. The Medtronic Symplicity Spyral system deliberately uses a "non-continuous" circumferential ablation pattern to prevent continuous vessel injury, which might lead to renal artery stenosis. The authors should provide histological evidence demonstrating the safety of continuous circumferential ablations achieved by the Netrod before asserting its superiority over non-continuous circumferential ablation in terms of efficacy or safety.

Missing Table: Table 3 is referenced in Results 3.1 but is not present in the manuscript.

Incomplete Information in Figure 9: The vessel diameter in the simulation model should be specified, as this parameter significantly influences the continuity of ablation lesions across different systems.

Inequitable System Comparisons: The default ablation times for different renal denervation systems are either 60 or 120 seconds. Comparing ablation extent at 30 seconds (Figure 8) or 5 seconds (Figure 9) provides an unfair comparison. Results should be shown for at least 60-second ablations across all systems.

Contradictory Results: In Results 3.3, the authors note that phantom experiments allow for optimal manual positioning of the Netrod™ Six-electrode against the vessel wall, while simulation experiments show poorer wall apposition. However, the smaller ablation depth and extent observed in phantom experiments contradict the expected benefits of better contact. This discrepancy requires explanation.

Incomplete Methodology: The ablation settings for other renal denervation systems are not described. These parameters should be clearly detailed to substantiate the fairness of the comparisons.

Reviewer #2: Comments for the Editors

The title of the manuscript does not indicate that this is an experiment and nor a clinical study.

This is not even an animal study, but a mathematical simulation on software and phantoms that limits the overall relevance of the experiments.

This should be specified in the title and the conclusions of the manuscript.

Thank you for the possibility of cooperating with the Journal, and kind regards.

PONE D-24-46135

Article type: Research Article

Study on the efficacy and safety of Netrod TM Six-electrode Radiofrequency Renal Denervation system on renal artery vessels

Comments to the authors

General

First of all, this experimental study has not been performed in renal arteries, but in a cylindrical synthetic mold made of Polyacrylamide. Therefore, the title should not mention the words renal artery vessels but rather “polyacrylamide phantom”. Likewise, not been performed in biological tissues of living organisms it cannot refer to “safety”.

Introduction. Authors should remove the comment on Flex and limit to spiral since the flex has no possibilities of spatial orientation of the electrode.

Specific

The following sentenced is not supported by the presented data and should be deleted:

“The NetrodTM Six-electrode device demonstrated effective ablation of renal sympathetic nerves, achieving temperatures of up to 45°C at distances of approximately 5.08 mm (>4 mm) from the electrodes. This confirms the feasibility and efficacy of using the NetrodTM Six-electrode for radiofrequency ablation of the renal sympathetic nerve”.

Indeed, such a sentence cannot be accepted since by no means the authors are testing the effect of the device on renal nerves, but in a phantom. The reader would be mislead with such a sentence that should clearly state that was changes color as an indirect indicator of heat is a synthetic material and not real tissues.

To affirm that the observed changes obtained after heating the RF electrodes are effective in ablating nerves, the experiments should be performed in animal models or in human cadavers.

Similarly, further in the results: “NetrodTM Six-electrode to be positioned optimally against the vessel wall”

Or in the discussion: “Results show that the NetrodTM Six- electrode system provides more comprehensive coverage and better continuity of the ablation zone compared to both the single electrode and the spiral six electrodes under the same setup. Lesion formation is influenced by several factors, including ablation power, ablation time, the number of electrodes, the area in contact with tissue, and blood flow”

Sentences as such cannot be proposed since there is no vessel wall, but a simple synthetic cylinder. Authors should therefore be clear with the reader and avoid any reference to tissues, vessels, blood and nerves.

Comparison of ablation effect of NetrodTM Six-electrode with single electrode and spiral six electrode: the authors do not explain the method used to detect and quantify the dispersion of the heath in figures 7,8 and 9.

What is the model used to test 6 spiral electrodes? The commercially available spiral system (Medtronic) has 4 electrodes and not 6. If the spiral 6 electrodes used in this study was developed by the investigators please provide details of the model used and clarify that is not the one commercially available, and that the effect described for 6 electrodes may not apply to a 4 electrodes spiral catheter.

Figure 10. Please explain how the phantom model was constructed and how was the effect of RF assessed.

The authors affirm that “The electrodes are treated as uniformly heated sources, spreading heat consistently into the bloodstream and surrounding tissue through conduction” how can they say that if their model has no fluid circulation, and therefore, temperature dispersion by blood temperature and flow cannot be calculated?

Authors used 60°C for 120 seconds with their device.

Other commercially available use variable temperature for 60 seconds. The authors should explain on which observations (ideally obtained in animals) they base the selection of temperature and duration of the energy delivery, and demonstrate in biological tissue of animals or cadavers, that such combination of temperature and duration is not creating unintended tissue damage to neighbor structures such as the artery wall, veins, lymphatic or urinary structures.

6. PLOS authors have the option to publish the peer review history of their article (what does this mean? ). If published, this will include your full peer review and any attached files.

**Do you want your identity to be public for this peer review?** For information about this choice, including consent withdrawal, please see our Privacy Policy .

Reviewer #1: No

Reviewer #2: No

---

## [Author Response · Author response to Decision Letter 1]

14 Jan 2025

Journal Requirements:

 Response: I have changed the format.

2.Please note that PLOS ONE has specific guidelines on code sharing for submissions in which author-generated code underpins the findings in the manuscript. In these cases, we expect all author-generated code to be made available without restrictions upon publication of the work.

Response:Yes, code will be made available upon publication of the work.

3.Thank you for stating the following financial disclosure:"This work was supported by the National Natural Science Foundation of China [31771021, 11832003]."Please state what role the funders took in the study.  If the funders had no role, please state: ""The funders had no role in study design, data collection and analysis, decision to publish, or preparation of the manuscript."

Response: This work was supported by the National Natural Science Foundation of China [31771021, 11832003]. The funder had role in study design, data collection and analysis.

 Response: Data Availability Statement is revised as follows: All relevant data are within the manuscript and its Supporting Information files.

 Response: I have modified the abstract to be consistent .

6. Please ensure that you refer to Figure 5 in your text as, if accepted, production will need this reference to link the reader to the figure.

 Response: Figure 5 was drawn by us.

Additional Editor Comments:

Additionally, in the first paragraph of Introduction, I would mention the clinical utility of renal denervation currently, beyond hypertension. You could refer to reviews encapsulating these findings, such as: https://doi.org/10.1016/j.cpcardiol.2023.102196.

Response: Thanks for the heads up on that.I added to the introduction the current state of research in hypertension other other clinical effects of renal denervation, etc.And cited the article as reference [11].

Content: Other studies[11] have also shown that RDN has the potential to extend to other autonomic nervous system disorders such as atrial fibrillation[12,13], heart failure[14,15] and tachycardia[16].

Furthermore, the discussion requires work. An additional paragraph on comparing and contrasting your findings with existing literature, what's novel about your results, and some background with pros and cons on all three types of electrodes used and their comparisons in either the introduction or second paragraph of the discussion is warranted.

Response: Thank for your advice, I have made overall revisions to the content of the article. I focus the article on the simulation and experimental data of NetrodTM Six-electrode to verify the safety and effectiveness of this electrode through animal experimental data, and no longer compare the advantages and disadvantages with other types. Therefore, I reduced the description of other types of electrodes.

Review Comments to the Author

Reviewer 1:

1.In this manuscript, the authors examined the depth and circumference of ablation lesions produced by the Netrod and other radiofrequency renal denervation systems through simulation models and phantom experiments. While the information provided has clinical significance, there are several methodological limitations and content inconsistencies that cast doubt on the conclusions.

Response: Thank for your advice, in general I modified my research to focus on the experiments on this electrode. In this article, I have further refined and perfected the data from the animal study, and comprehensively demonstrated the efficacy and safety of the NetrodTM Six-electrode by using the data obtained from simulation, phantom experiments and the animal study. For details of the animal study, please refer to the section labeled “Animal study” (highlighted in red here), which records and analyzes the data (blood pressure changes and histological analysis) from the animal study in detail.

2.The title (referring to efficacy and safety, as shown on the first page, which differs from the title on the first page of the text) is inconsistent with the content presented, which only addresses the extent of ablations and does not discuss efficacy or safety.

Major limitations requiring clarification:

Response: Thank for your suggestion to add animal testing to verify its safety and efficacy. In the course of the experiment, through the monitoring of the animals' blood pressure, the stable performance of the NetrodTM Six-electrode in practical application was clearly demonstrated. Specifically, the experimental animals did not experience any abnormal physiological reactions or adverse reactions after receiving the electrode treatment, and all vital signs remained stable, which fully demonstrates that its safety can stand the test.The details are as follows:

Content: During the postoperative observation period, the experimental animals did not show any abnormalities related to the treatment. No abnormalities such as stenosis or dissection were observed in the renal arteries of the animals when the angiography was performed before dissection, and no abnormalities were observed when the animals were dissected for observation and sampling. The above experiments prove the safety of the device.

3.Abstract Inconsistency: The abstract on the first page of the PDF file mentions animal studies and histological analysis. However, the main text and its abstract only present data on ablation lesion depth and circumference from different radiofrequency renal denervation systems. Additionally, the titles differ between sections. The content is limited and inconsistent with the title on the first page ("Study on the efficacy and safety of Netrod™ Six-electrode Radiofrequency Renal Denervation system on renal artery vessels"), as no information regarding clinical efficacy and safety is presented.

Response:Thank for your advice. I have refined the animal studies and histological analysis in the main text to validate the study on the efficacy of NetrodTM Six-electrode Radiofrequency through blood pressure changes in the animals and The results of H&E staining of the ablation sites. Renal Denervation system so that the abstract corresponds to the main text.At the same time I improved some of the article's headings to align them with the body content.

4..Unsupported Assumptions in Introduction: The authors assume that "continuous circumferential (ring)" ablation is superior for renal denervation without providing supporting evidence. The Medtronic Symplicity Spyral system deliberately uses a "non-continuous" circumferential ablation pattern to prevent continuous vessel injury, which might lead to renal artery stenosis. The authors should provide histological evidence demonstrating the safety of continuous circumferential ablations achieved by the Netrod before asserting its superiority over non-continuous circumferential ablation in terms of efficacy or safety.

Response:As the reviewer noted, we do not discount the safety of Medtronic products in terms of continuous ablation, but only assessed the safety of continuous ablation with our own ablation electrodes in 18 animals. As a measure of efficacy, blood pressure was measured at regular intervals before and during the post-test observation period, and we measured the change in blood pressure at three points of time after RDN. During survival, animals were observed clinically. To investigate the short- term, medium- term, and long-term safety of the test article, three points of time, 7, 30, and 60 days, were included in this experiment. On postoperative days 7 (±1), 30 (±3) and 60 (±5) (with the day of surgery as 0 day), renal arteriograms were performed in animals scheduled for dissection to visualize the test site and renal arteries with nerves were harvested for immuno-histochemical evaluation. Quantification of protein denaturation and necrosis of renal sympathetic nerves around renal arteries were using hematoxylin-eosin (H&E) staining. Specific experimental results are shown in Animal experiment results. We believe that the design of the net basket electrodes allows multiple electrodes to contact the vessel simultaneously and create a continuous line of ablation at the desired location, which can improve the therapeutic efficacy.

5.Missing Table: Table 3 is referenced in Results 3.1 but is not present in the manuscript.

Response:Yes, thanks for the advice. The tables have been switched in content and location, and all tables have been sorted.

6.Incomplete Information in Figure 9: The vessel diameter in the simulation model should be specified, as this parameter significantly influences the continuity of ablation lesions across different systems.

Response:Yes, thanks for the advice. Figure 9 was removed because the focus was no longer on the helical electrode approach data. The diameter of the vessels in the simulation and in the soma experiments was 6 mm.There is a reference in “Finite element method modeling”:The tissue included two major parts, the renal artery�6mm diameter).

7.Inequitable System Comparisons: The default ablation times for different renal denervation systems are either 60 or 120 seconds. Comparing ablation extent at 30 seconds (Figure 8) or 5 seconds (Figure 9) provides an unfair comparison. Results should be shown for at least 60-second ablations across all systems.

Response:Yes, thanks for the advice. I agree with the reviewers that there was unfairness in the way the previous comparisons with other systems were made. Therefore, in this revision of the paper, instead of comparing other systems to highlight the advantages of the NetrodTM Six-electrode, we will focus on the presentation of the experimental data of the system itself. We will analyze the data obtained from simulations and in vivo experiments, and through detailed interpretation and analysis of these data, we will strongly argue efficacy and safety of the NetrodTM Six-electrode.

8.Contradictory Results: In Results 3.3, the authors note that phantom experiments allow for optimal manual positioning of the Netrod™ Six-electrode against the vessel wall, while simulation experiments show poorer wall apposition. However, the smaller ablation depth and extent observed in phantom experiments contradict the expected benefits of better contact. This discrepancy requires explanation.

Response:Thanks for the advice. It may be that our expression is not precise enough, we think: the simulation considers rigid walls or does not consider deformation, so the fit is worse than that of the phantom experiments , and therefore the depth is smaller. During the actual operation of the phantom experiments , adjustments can be made manually to make the mesh basket fit tightly, so the fit is better in phantom experiments, and the ablation depth is greater This may be the reason for the difference.

9.Incomplete Methodology: The ablation settings for other renal denervation systems are not described. These parameters should be clearly detailed to substantiate the fairness of the comparisons.

Response:Thanks for the advice. I agree with the reviewers that the previous comparisons with other systems were indeed unfair. Therefore, in this article, we will no longer use the comparison with other systems to highlight the advantages of theNetrodTM Six-electrode, but focus on the results of the experimental data of this system, with the help of concrete experimental data to strongly argue the rationality and effectiveness of this system, so that its advantages can be presented visually and more convincing.

Reviewer 2:

1.The title of the manuscript does not indicate that this is an experiment and nor a clinical study.This is not even an animal study, but a mathematical simulation on software and phantoms that limits the overall relevance of the experiments.This should be specified in the title and the conclusions of the manuscript.

Response:Yes, thank for your advice, and as a whole I've revised my research. The article was refined, mainly because the previous study was biased towards the mathematical simulation of the simulation and phantom experiments, but because of the lack of experimental data from animals or human beings, it limited the overall relevance of the study and did not well reflect the safety feasibility of the studied system, so the study increased the animal studies to further improve the experimental data on the basis of the mathematical simulation and match the simulation data to improve the study of this system.

2.First of all, this experimental study has not been performed in renal arteries, but in a cylindrical synthetic mold made of Polyacrylamide. Therefore, the title should not mention the words renal artery vessels but rather “polyacrylamide phantom”. Likewise, not been performed in biological tissues of living organisms it cannot refer to “safety”.Introduction. Authors should remove the comment on Flex and limit to spiral since the flex has no possibilities of spatial orientation of the electrode.

Response:Yes, thanks for the advice. I added animal testing to the study to demonstrate “safety” through changes in animal blood pressure data and histology.We did animal experiments, and the significant decrease in blood pressure in the animals after surgery and the histologic data after autopsy verified the effectiveness and safety (see Animal experiment results for details). And remove the comment on Flex and limit to spiral since the flex has no possibilities of spatial orientation of the electrode.

3.The NetrodTM Six-electrode device demonstrated effective ablation of renal sympathetic nerves, achieving temperatures of up to 45°C at distances of approximately 5.08 mm (>4 mm) from the electrodes. This confirms the feasibility and efficacy of using the NetrodTM Six-electrode for radiofrequency ablation of the renal sympathetic nerve”.Indeed, such a sentence cannot be accepted since by no means the authors are testing the effect of the device on renal nerves, but in a phantom. The reader would be mislead with such a sentence that should clearly state that was changes color as an indirect indicator of heat is a synthetic material and not real tissues.

Response:Yes, thanks for the advice. I think the reviewer made a reasonable point, and this formulation has been deleted, and to add in Phantom experiment validation:”The body mold changes color from blue to white when the temperature exceeds 45°C.”

4.To affirm that the observed changes obtained after heating the RF electrodes are effective in ablating nerves, the experiments should be performed in animal models or in human cadavers.

Response:Yes, thanks for the advice. The study has added animal experiments. Comparisons were made with the simulation to illustrate the accuracy of the simulation. Also blood pressure changes by animal experiments and renal arteriography was performed on animals scheduled for dissection to visualize the test site and renal arteries with nerves were collected for immunohistochemical evaluation.

5.Similarly, further in the results: “NetrodTM Six-electrode to be positioned optimally against the vessel wall”Or in the discussion: “Results show that the NetrodTM Six- electrode system provides more comprehensive coverage and better continuity of the ablation zone compared to both the single electrode and the spiral six electrodes under the same setup. Lesion formation is influenced by several factors, including ablation power, ablation time, the number of electrodes, the area in contact with tissue, and blood flow”

Sentences as such cannot be proposed since there is no vessel wall, but a simple synthetic cylinder. Authors should therefore be clear with the reader and avoid any reference to tissues, vessels, blood and nerves.

Response:Yes, thank you for your suggestion. Instead of comparing the ablation zones of the two, we did a complete simulation, ex vivo and animal experiments for this electrode. So this part ha

---

## [Decision Letter · Decision Letter 1]

2 Feb 2025

PONE-D-24-46135R1Study on the efficacy of NetrodTM Six-electrode Radiofrequency Renal Denervation system on renal artery vesselsPLOS ONE

Dear Dr. Nan,

Thank you for submitting your manuscript to PLOS ONE. After careful consideration, we feel that it has merit but does not fully meet PLOS ONE’s publication criteria as it currently stands. Therefore, we invite you to submit a revised version of the manuscript that addresses the points raised during the review process.

Please look below for changes suggested by me, the Editor, and the reviewers at the end of this email.

We look forward to receiving your revised manuscript.

Kind regards,

Aman Goyal

Academic Editor

PLOS ONE

Journal Requirements:

Additional Editor Comments:

Dear Dr. Nan,

As I mentioned during the round one of revision. Please do add a more complete and expanded discussion. The discussion is too short and does not delve into comparing of your own results, with the results of existing literature on this topic that is published.

Please follow this format:

First paragraph of discussion: Key results

Second paragraph: Importance of your findings, and how it is clinically relevant.

Third paragraph: Comparison of your results with existing literature.

Fourth paragraph: Strengths and Limitations of your study

Followed by conclusion.

Please make sure the discussion has points that bring to light the clinical relevance of your study, how it guides future research that needs to be conducted.

Please address points raised by the reviewers as well.

Reviewers' comments:

Reviewer's Responses to Questions

**Comments to the Author**

1. If the authors have adequately addressed your comments raised in a previous round of review and you feel that this manuscript is now acceptable for publication, you may indicate that here to bypass the “Comments to the Author” section, enter your conflict of interest statement in the “Confidential to Editor” section, and submit your "Accept" recommendation.

Reviewer #1: All comments have been addressed

Reviewer #2: (No Response)

2. Is the manuscript technically sound, and do the data support the conclusions?

Reviewer #1: Partly

Reviewer #2: (No Response)

3. Has the statistical analysis been performed appropriately and rigorously? 

Reviewer #1: Yes

Reviewer #2: (No Response)

4. Have the authors made all data underlying the findings in their manuscript fully available?

Reviewer #1: No

Reviewer #2: (No Response)

5. Is the manuscript presented in an intelligible fashion and written in standard English?

Reviewer #1: No

Reviewer #2: (No Response)

6. Review Comments to the Author

Reviewer #1: The authors revised extensively according to the comments. The revised version is much improved, though there are still some points need to be clarified. There are quite a few grammatical/tense errors, which require detailed English editing to make it friendly to the readers.

1. Title: The title “Study on the efficacy of NetrodTM Six-electrode Radiofrequency Renal Denervation system on renal artery vessels” does not match the findings. The authors provide information includes studies based on mathematical modeling, phantom experiments and animal study. It is not solely based on studies on renal arteries. It would be more rational and informative to revise the title to “The efficacy of NetrodTM Six-electrode Radiofrequency Renal Denervation system: mathematical modeling, phantom experiments and animal studies”.

2. Abstract: The statement “Systematic ablation can reach the target temperature in a shorter period of time, with faster warming, greater peripheral coverage of the lesion, and a more continuous ablation zone.” is not consistent with the findings. Given there is no comparison with other devices, it would not be rational to state that Netrod is “shorter”, “faster”, or “greater”. The authors should revise and describe the findings in a neutral manner.

3. Lesion continuity: Given that the depth of 45 degree boundary at 30 seconds of heating was 2.69 mm (Table 3), the distance between electrodes should be less than 5.4 mm to achieve continuity. The authors should introduce the distance between electrodes in the manuscript to make readers realize how continuity could be established, rather than simply showing a figure with no supportive data. Further, the authors did not provide data regarding the statement in the Abstract “Circumferential coverage was up to 91.99% and the ablation zone was continuously.” The authors should provide relevant data.

4. Lesion depth and width in animal studies: According to Figure 12, the mean ablation depth in animal studies seems to be less than 3 mm, which is less than the distance reported in simulation and phantom studies (5-6 mm). As the authors mentioned in the Discussion “AI Raisi et al. [30] reported a circumferential coverage angle of 54.90° ± 4.36° for a single electrode RF application, encompassing approximately 15% of the arterial wall circumference”, the authors should provide exact numbers of ablation depth and width (or coverage angle) in a Table and a representative H&E figure to show how the lesion depth and width were measured. The authors should discuss why there was such a discrepancy in ablation depth between animal studies and simulation studies and whether it would impact the so-called “continuity” to be established and the corresponding ablation strategy.

5. Confusing terms and grammatical errors: It is confusing what “test article”, “preanatomic” (Figure 7 legend), “blank control” refer to. It might be “treatment group”, “pre-ablation” and “sham-control”, but it is definitely hard for readers. Extensive language editing is required.

Reviewer #2: The authors have aproperly addressed my requests.

I have no other comments

The authors have aproperly addressed my requests.

I have no other comments

7. PLOS authors have the option to publish the peer review history of their article (what does this mean? ). If published, this will include your full peer review and any attached files.

**Do you want your identity to be public for this peer review?** For information about this choice, including consent withdrawal, please see our Privacy Policy .

Reviewer #1: No

Reviewer #2: No

---

## [Author Response · Author response to Decision Letter 2]

14 Mar 2025

Journal Requirements:

Response: Incomplete display of page numbers for reference[1] and incomplete journal titles for reference[3]. I've modified it.

Additional Editor Comments:

As I mentioned during the round one of revision. Please do add a more complete and expanded discussion. The discussion is too short and does not delve into comparing of your own results, with the results of existing literature on this topic that is published.

As I mentioned during the round one of revision. Please do add a more complete and expanded discussion. The discussion is too short and does not delve into comparing of your own results, with the results of existing literature on this topic that is published.

Please follow this format:

First paragraph of discussion: Key results

Second paragraph: Importance of your findings, and how it is clinically relevant.

Third paragraph: Comparison of your results with existing literature.

Fourth paragraph: Strengths and Limitations of your study

Response: Thank you for your suggestion. I have added the discussion section, which has been labeled in the original article.

Review Comments to the Author

Reviewer 1:

1.Title: The title “Study on the efficacy of NetrodTM Six-electrode Radiofrequency Renal Denervation system on renal artery vessels” does not match the findings. The authors provide information includes studies based on mathematical modeling, phantom experiments and animal study. It is not solely based on studies on renal arteries. It would be more rational and informative to revise the title to “The efficacy of NetrodTM Six-electrode Radiofrequency Renal Denervation system: mathematical modeling, phantom experiments and animal studies”..

Response: Thank you very much for your advice. I have revised the title.

2.Abstract: The statement “Systematic ablation can reach the target temperature in a shorter period of time, with faster warming, greater peripheral coverage of the lesion, and a more continuous ablation zone.” is not consistent with the findings. Given there is no comparison with other devices, it would not be rational to state that Netrod is “shorter”, “faster”, or “greater”. The authors should revise and describe the findings in a neutral manner.

Response: Thank you very much for your suggestion. In the absence of comparative data, it should not be generalized that these results are superior or inferior to other equipment.

3.Lesion continuity: Given that the depth of 45 degree boundary at 30 seconds of heating was 2.69 mm (Table 3), the distance between electrodes should be less than 5.4 mm to achieve continuity. The authors should introduce the distance between electrodes in the manuscript to make readers realize how continuity could be established, rather than simply showing a figure with no supportive data. Further, the authors did not provide data regarding the statement in the Abstract “Circumferential coverage was up to 91.99% and the ablation zone was continuously.” The authors should provide relevant data.

Response: Thank you very much for your advice. It is mentioned in the manuscript that based on the construction of the mesh basket-like electrodes of this system, the distance between the electrodes can be flexibly varied with the diameter of the blood vessels, and the distance between the electrodes is not a fixed value, so the distance between the electrodes is not explicitly mentioned in the manuscript. For continuity results, see the following and Figures 9 and 10.

Content: Each of the NetrodTM Six-electrode sits on a separate mesh basket of wires, which better conforms to the vessel wall as the lumen changes, keeping the electrode arrangement consistent.

The continuity of the NetrodTM Six-electrode ablation zone at an ablation time of 30s is shown in Fig 9. Each of the NetrodTM Six-electrode sits on a separate mesh basket of wires, which better conforms to the vessel wall as the lumen changes, keeping the electrode arrangement consistent.

As shown in Fig 10, during the ablation process, the six electrodes discharge to form a continuous 360º spiral ribbon, with the ablation ranges of the first and sixth electrodes overlapping radially.

4.Lesion depth and width in animal studies: According to Figure 12, the mean ablation depth in animal studies seems to be less than 3 mm, which is less than the distance reported in simulation and phantom studies (5-6 mm). As the authors mentioned in the Discussion “AI Raisi et al. [30] reported a circumferential coverage angle of 54.90° ± 4.36° for a single electrode RF application, encompassing approximately 15% of the arterial wall circumference”, the authors should provide exact numbers of ablation depth and width (or coverage angle) in a Table and a representative H&E figure to show how the lesion depth and width were measured. The authors should discuss why there was such a discrepancy in ablation depth between animal studies and simulation studies and whether it would impact the so-called “continuity” to be established and the corresponding ablation strategy.

Response: Fig. 13 replaces Fig. 12, and the data in Fig. 13 are average data, some of which can reach the distance reported in the simulation and modeling studies (5-6 mm) because the data in the simulation and modeling studies were set up for an ideal state, whereas there is a gap between the depth of the simulation and the animal experiments due to the real arterial structure and arterial shape of the animals in the animal experiments. For the H&E graphs of the animal experiments, the initial intention in this study was to determine the effectiveness of this system through simulation, and to verify the safety of ablation through H&E staining of the hypertensive canine model, so the animal experiments did not go to validate the data of the simulation, but to further study that this system is safe, so it wasn't focused on, but the next step will be to go to get the accurate data through more animal experiments .

The discussion of animal experiments has been added as follows:

In this study, we verify the effectiveness of this system through modeling studies, and animal experiments are further to go to verify the effectiveness and safety of this system, as the data in the modeling studies are set up for an ideal state, and there is a gap between the depth and the animal experiments due to the real arterial structure of the animals and the shape of the arteries in the animal experiments.

5.Confusing terms and grammatical errors: It is confusing what “test article”, “preanatomic” (Figure 7 legend), “blank control” refer to. It might be “treatment group”, “pre-ablation” and “sham-control”, but it is definitely hard for readers. Extensive language editing is required.

Response: Thanks for the advice. The “test article” refers to “the experimental dogs of treatment group”. The “preanatomic” refer to “pre-dissection”. The “blank control” refer to “control group”.

I replaced “test article” with “the experimental dogs of treatment group” and replaced ”blank control” with ” control”.

---

## [Decision Letter · Decision Letter 2]

6 Apr 2025

The efficacy of NetrodTM Six-electrode Radiofrequency Renal Denervation system: mathematical modeling, phantom experiments and animal studies

PONE-D-24-46135R2

Dear Dr. Nan,

We’re pleased to inform you that your manuscript has been judged scientifically suitable for publication and will be formally accepted for publication once it meets all outstanding technical requirements.

Kind regards,

Aman Goyal, MD

Academic Editor

PLOS ONE

Additional Editor Comments (optional):

Reviewers' comments:

Reviewer's Responses to Questions

**Comments to the Author**

1. If the authors have adequately addressed your comments raised in a previous round of review and you feel that this manuscript is now acceptable for publication, you may indicate that here to bypass the “Comments to the Author” section, enter your conflict of interest statement in the “Confidential to Editor” section, and submit your "Accept" recommendation.

Reviewer #1: All comments have been addressed

2. Is the manuscript technically sound, and do the data support the conclusions?

Reviewer #1: Yes

3. Has the statistical analysis been performed appropriately and rigorously? 

Reviewer #1: Yes

4. Have the authors made all data underlying the findings in their manuscript fully available?

Reviewer #1: Yes

5. Is the manuscript presented in an intelligible fashion and written in standard English?

Reviewer #1: Yes

6. Review Comments to the Author

Reviewer #1: In this revised manuscript, the authors adequately responded to the comments I made in the review. There are some minor issues to be solved in this version, which I mention below.

1. Depth of ablation in animal studies: It should be commended that the authors displayed the depth of ablation in detail in Figure 13. Even though the depth is smaller than what had been observed in simulation and phantom studies, the authors should emphasize in the Discussion that the depth measured in animal studies was based on the furthest nerve damaged, rather than soft tissue involved, which would be difficult to judge. Hence, the depth of ablation could be underestimated inherently, if the distribution of nerves surrounding the renal artery segment was scarce beyond 3 mm. The authors should mention this as one of the reasons why there were discrepancies between the depths observed in different settings.

2. Abstract: The timing of assessment (7, 30, and 60 days) in animal studies should be mentioned in the Abstract.

3. Abstract: The statement should be neutral. “Strong” in the last sentence of the Abstract seems not appropriate.

4. “TM” should be typed as superscript throughout the article.

5. Results: The sentence in Animal Study Results, “pre-anatomic diastolic and mean arterial pressures in animals 101 may not be statistically different relative to pre-ablation due to individual differences (P>0.05)”, is confusing. The word “pre-anatomic” should be “post-ablation”.

7. PLOS authors have the option to publish the peer review history of their article (what does this mean? ). If published, this will include your full peer review and any attached files.

**Do you want your identity to be public for this peer review?** For information about this choice, including consent withdrawal, please see our Privacy Policy .

Reviewer #1: **Yes: ** Tzung-Dau Wang

---

## [Editor Report · Acceptance letter]

PONE-D-24-46135R2

PLOS ONE

Dear Dr. Nan,

I'm pleased to inform you that your manuscript has been deemed suitable for publication in PLOS ONE. Congratulations! Your manuscript is now being handed over to our production team.

Kind regards,

on behalf of

Dr. Aman Goyal

Academic Editor

PLOS ONE